# Properties of Cement Mortar Using Limestone Sludge Powder Modified with Recycled Acetic Acid

**Hwa-Sung Ryu [1], Deuck-Mo Kim [1], Sang-Heon Shin [1], Wan-Ki Kim [2], Seung-Min Lim [3]** and **Won-Jun Park [4],***

[1] Hanyang Experiment and Consulting Co., Hanyang University, Ansan 15588, South Korea; rhsung73@hanyang.ac.kr (H.-S.R.); golanhae@naver.com (D.-M.K.); monanom80@gmail.com (S.-H.S.)

[2] Department of Architectural Engineering, Hyupsung University, Hwaseong, 445-745, South Korea; wankkim@uhs.ac.kr

[3] Innovative Durable Building and Infrastructure Research Center, Hanyang University, Ansan 15588, South Korea; smlim09@hanyang.ac.kr

[4] Department of Architectural Engineering, Kangwon National University, Samcheok 25913, South Korea

* Correspondence: wjpark@kangwon.ac.kr; Tel.: +82-33-570-9529

**Abstract:** One of the various methods of manufacturing low-carbon cement is substituting limestone powder as a raw material or admixture. Limestone sludge powder (LSSP) has the same composition as that of limestone powder. The surface characteristics of LSSP powder modified with recycled acetic acid (RAA) and the characteristics of cement using this modified LSSP as a substitute were investigated in this study. The surface of LSSP modified with RAA was converted into calcium acetate and had a large grain size. When conventional LSSP was used as a substitute for cement, the initial strength increased owing to improved pore filling; however, the strength after 28 days of aging was lower than that of non-substituted cement. In the case of modified LSSP being replaced with cement at up to 10% of the cement weight, however, the calcium acetate on its surface increased the amount of hydration products in the cement, thereby increasing both the initial and the long-term strength.

**Keywords:** cement mortar; limestone sludge; recycled acetic acid; surface modification

## 1. Introduction

One of the major issues in the construction industry in recent years has been the demand for cement materials that can reduce carbon emissions. The reality is that 1 t of $CO_2$ is generated to produce 1 t of cement, and most studies have focused on reducing cement usage or using cement substitutes [1,2]. One of the most widely used materials in the cement production process is limestone powder. In Europe, limestone powder may be used at up to 35% according to the EN 197-1 standard, whereas in Canada and the United States, up to 5% Portland cement and 15% Portland limestone cement may be used [3–5]. Limestone powder contains a large amount of calcium carbonate. Calcium carbonate and silica or nanocarbon are used as fillers in organic and inorganic complexes. These fillers serve to improve the viscosity or physical performance of polydimethylsiloxane (PDMS) [6–10], which is used for penetrating water repellency and surface penetration coating. The functional groups on the surfaces of these fillers may be changed to improve their reactivity with the polymer or to optimize the dispersion ability. Compared with silica or nanocarbon, calcium carbonate is a less expensive functional filler [8–15].

In the steel industry, burnt lime is produced by washing and burning limestone. The sludge composed of powders washed from limestone is known as limestone sludge powder (LSSP). LSSP has the same composition as limestone powder and can be used as a substitute for cement and, therefore,

as an admixture material of cement mortars. Limestone powder is a raw material of cement, and the LSSP used in this study is an admixture for mixing with cement to reduce cement use. LSSP has the same composition as that of limestone powder and can be replaced with cement as an admixture material. The shape and particle size of limestone powder are relatively good compared to those of cement particles. Replacing a portion of cement with limestone powder improves the fluidity of the mortar [16,17]. It has been reported that it also contributes to quality enhancements such as improved resistance to material separation and enhanced strength, owing to the pore-filling effect. Studies have suggested that limestone powder accelerates the hydration of alite in cement. However, calcium carbonate, which is a major component of limestone powder, is mostly non-hydrated and, thus, it is possible that the initial strength can be lowered when a substantial amount of LSSP is used [16–21].

Similarly, using fly ash and blast furnace slag, which are mixed materials used to improve the performance of cement, may delay the hydration of cement and lower its initial strength. To induce initial hydration early, researchers have used strengthening agents such as compounds containing carboxyl groups, lithium compounds, and calcium nitrate [22–29]. Waste acetic acid, a byproduct of liquid crystal display (LCD) surface etching processes, has a low concentration and is restricted to reuse in industry. The calcium acetate produced by the reaction of calcium salts with acetic acid can promote the hydration of cement owing to the carboxyl groups of this acid. Calcium acetate also improves the initial strength and reduces internal voids. Hence, modification of the surfaces of limestone powders using acetic acid can improve the early strength properties of these powders when they are used as admixtures for cement.

Therefore, in this study, an admixture material was developed by modifying LSSP using acetic acid to form reactive limestone powders to enhance the performance of cement mortar. The properties of the samples were investigated in terms of mortar setting time and compressive strength. They were also analyzed using X-ray diffraction (XRD), thermogravimetric/differential thermal analysis (TG-DTA), and scanning electron microscopy (SEM).

## 2. Materials and Methods

### 2.1. Materials

Table 1 shows the chemical properties of the binders used in the experiments. The cement used was an ordinary type 1 Portland cement (OPC). Chemical analysis of LSSP showed that the CaO and MgO contents were 51.5% and 1.60%, respectively. This corresponds to about 90% of $CaCO_3$, which corresponds to high-grade limestone, and is similar in chemical properties to reagent-grade $CaCO_3$. Recycled acetic acid (RAA) was obtained after being extracted from the separation process of acids used in etching processes. For this study, RAA with 60% solid content was used. Figure 1 shows the Fourier transform infrared (FT-IR) spectroscopy results for the RAA. Organic acids are denoted by the peaks in the range of 3300–2500 $cm^{-1}$, where carboxylic groups are traditionally located. C=O groups (1760–1690 $cm^{-1}$) and C–O groups (1320–1210 $cm^{-1}$) are also present in this spectrum. Some examples of carboxylic acids are acetic acid, formic acid, glucosan, and propionic acid. The functional group of the RAA used for this study was confirmed to be carboxylic acid.

**Table 1.** Chemical composition of the ordinary type 1 Portland cement (OPC), $CaCO_3$, and limestone sludge powder (LSSP) (%).

| | CaO | SiO$_2$ | MgO | Al$_2$O$_3$ | SO$_3$ | Fe$_2$O$_3$ | K$_2$O | Na$_2$O | LOI |
|---|---|---|---|---|---|---|---|---|---|
| OPC | 63.35 | 21.09 | 3.32 | 4.34 | 3.09 | 2.39 | 1.13 | 0.29 | 1.0 |
| CaCO$_3$ | 52.53 | 1.18 | 2.47 | 0.47 | 0.02 | 0.43 | 0.13 | 0 | 42.77 |
| LSSP | 53.16 | 3.96 | 1.09 | 2.13 | 0.08 | 1.4 | 0.46 | 0.05 | 37.7 |

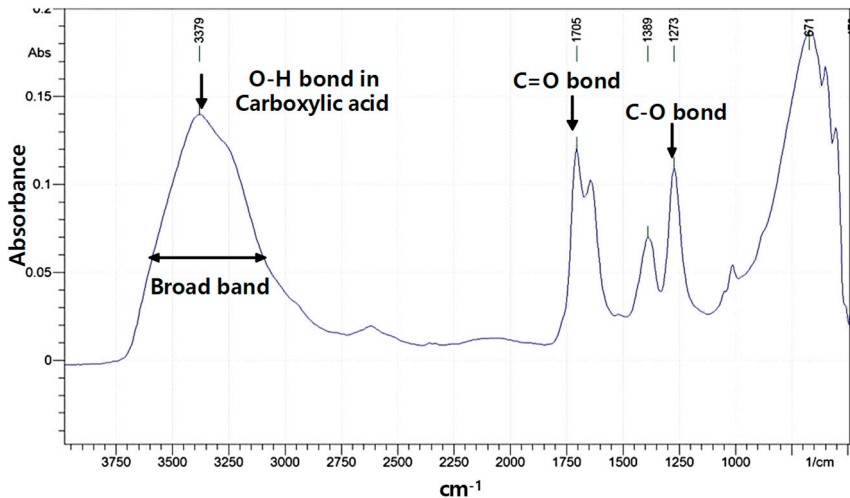

**Figure 1.** Fourier transform infrared (FT-IR) spectrum of recycled acetic acid (RAA).

## 2.2. Methods

Figure 2 shows the surface modification process of the LSS. A 10% aqueous solution was made using RAA with 60% solid content; then, 1 kg of the aqueous solution was mixed with 100 g of LSS and the mixture was kneaded for about 10 min. The modified calcium carbonate was collected through a filter. The reaction with acetic acid on the surface of the LSS is as follows:

$$2CH_3COOH + CaCO_3 = Ca (CH_3COO)_2 + H_2O + CO_2 \tag{1}$$

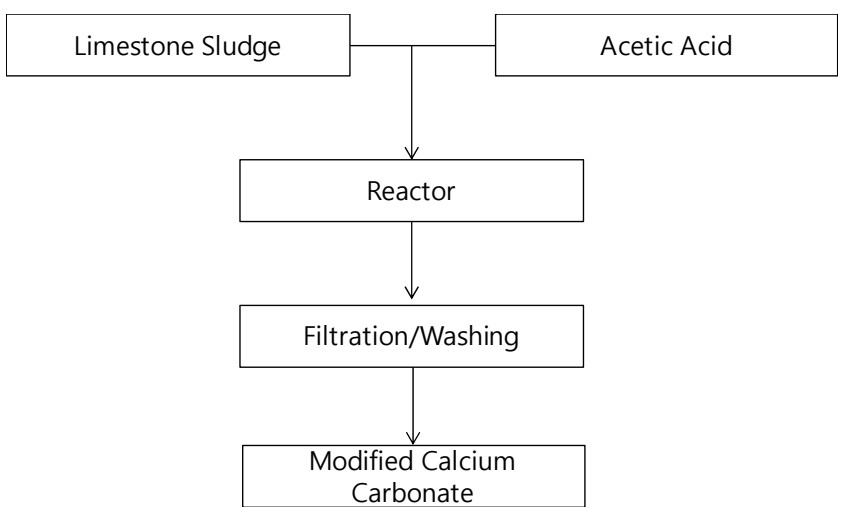

**Figure 2.** Flow diagram for the modified calcium carbonate.

Table 2 shows the mixing proportions of samples, where OPC was substituted by conventional and modified LSSP. The specimens for evaluating the characteristics of the cement mortar were prepared by mixing the mortar and placing it into a beam-shaped mold with dimensions of $40 \times 40 \times 160$ mm$^3$. The prepared specimens were cured for 18 h at room temperature and demolded for underwater curing. The compressive strength was then measured after different curing times: 18 h, 1 day, 3 days, 7 days, and 28 days, according to ISO 679 [30]. The mortar fluidity was measured according to ASTM C 1437. The mortar setting time was measured according to the initial and final setting time specified in ASTM C 191, and the mortar compressive strength was measured according to ASTM C 109.

**Table 2.** Mixing proportions (unit: g).

| Sample | OPC (Ordinary Portland Cement) | LSSP (Limestone Sludge Powder) | Modified LSSP | Silica sand | Water |
|---|---|---|---|---|---|
| OPC | 1000 | - | - | | |
| OPC-L5 | 950 | 50 | | | |
| OPC-L10 | 900 | 100 | | | |
| OPC-L15 | 850 | 150 | | | |
| OPC-L20 | 800 | 200 | | 2000 | 340 |
| OPC-ML5 | 950 | | 50 | | |
| OPC-ML10 | 900 | | 100 | | |
| OPC-ML15 | 850 | | 150 | | |
| OPC-ML 20 | 800 | | 200 | | |

The age of the sample for analysis is 28 days, and immersed in acetone for 24 hours to stop hydration of individual samples. The water was then removed in a 60 °C dryer. Sample preparation for each analysis is as follows. SEM samples were made of flake samples with a width of 0.5 cm × 0.5 cm and a thickness of 5 mm or less. TGA and XRD analysis specimens were prepared with paste, without aggregates, to suppress noise of sample. After grinding to below 100 microns at 28 days of age, it was immersed in an acetone solution and dried using a vacuum distillation apparatus.

## 3. Results and Discussion

### 3.1. Characteristics of Modified LSSP

Figure 3 shows the SEM images of the surfaces of conventional and modified LSSP. On the surface of the modified LSSP, calcium acetate crystals with relatively large grain sizes can be seen. This is because the calcium carbonate in powder form was dissolved by acetic acid. The pyrolysis characteristics of conventional LSSP showed a rapid weight loss of approximately 84% between 700 °C and 800 °C, as shown in Figure 4. The pyrolysis of the byproducts showed a heating curve with a weight loss of approximately 1.476% at around 350 °C and 59% between 700 °C and 800 °C. The modified LSSP showed about 30% weight loss at 800 °C. The result of the particle size distribution measurement of the LSS showed an average particle size of 18.5 μm. In the case of the modified LSSP, the average particle size was 17.2 μm.

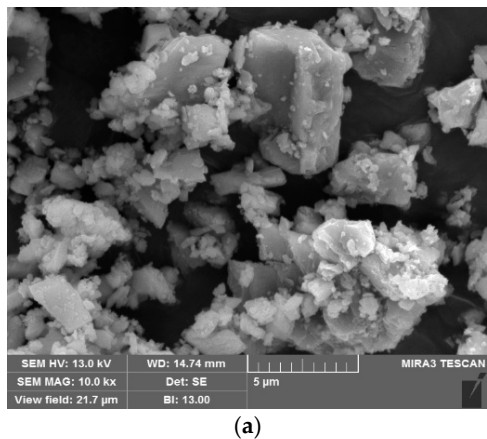
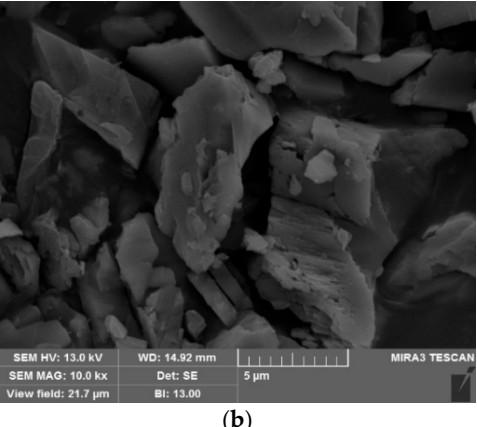

(**a**) (**b**)

**Figure 3.** Scanning electron microscopy (SEM) images of limestone sludge powder (LSSP): (**a**) conventional LSSP and (**b**) modified LSSP.

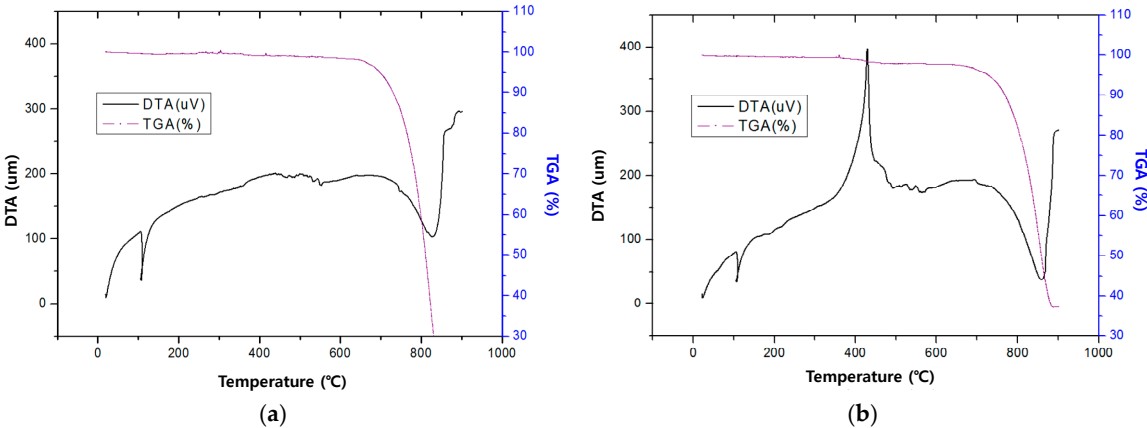

**Figure 4.** Results of the thermogravimetric/differential thermal analysis (TG-DTA): (**a**) conventional LSSP and (**b**) modified LSSP.

### 3.2. Mortar Application Characteristics

#### 3.2.1. Setting Time

Figure 5 shows the measured setting time of mortar mixed with conventional and modified LSSP. The use of conventional LSSP showed no noticeable difference in both the initial and the final setting time. In the case of the modified LSSP, however, the initial setting time decreased as the substitution ratio increased. The final setting time decreased by up to 10%, but increased for LSSP content above 15%. It is possible that the initial setting time decreased owing to hydration reactions induced by calcium acetate on the surface of the modified LSSP; however, the presence of excessive organic calcium for higher contents of LSSP caused the increase in values of the final setting time. Acetic acid, used for LSS modification, has been reported to increase the conductivity in the initial cement paste, thereby increasing the ionic concentration of $Ca^{2+}$ and $Al\,(OH)_4^{-}$. However, an excessive amount of acetic acid may cause hydration delay with increasing adsorption amount on the surface of cement paste particles [31].

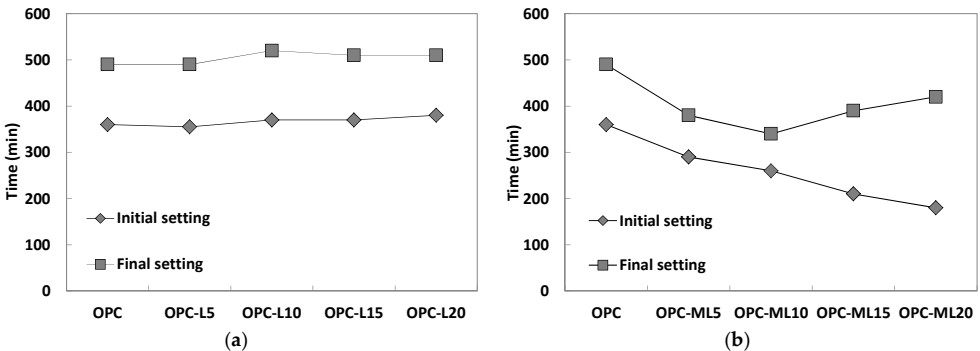

**Figure 5.** Setting time of cement mortar: (**a**) Mortar using conventional LSSP; (**b**) Mortar using modified LSSP.

#### 3.2.2. Compressive Strength

Figure 6a shows the compressive strength of the cement mortar according to the substitution ratio of conventional LSSP for different curing times. Overall, the compressive strength of the mortar was not significantly different from that of OPC, even when the replacement ratio of conventional LSSP increased at all curing ages. Figure 6b shows the graph of the compressive strength according to the substitution ratio of the modified LSSP. Substitution with the modified LSSP produced lower initial compressive strength than that of OPC for all specimens. After 1 day of aging, the substitution

proportions of 5% and 10% modified LSSP showed higher strength than that of OPC. After 28 days of aging, the 5% substitution resulted in strength superior to that of OPC. We believe that the introduction of modified LSSP into the space between cement particles improved pore filling and induced hydration reactions around the unreactive calcium carbonate particles, thereby increasing the strength. At a replacement rate of up to 15%, the mortar using LSSP and modified LSSP showed a performance equal to or higher than that of the mortar using OPC.

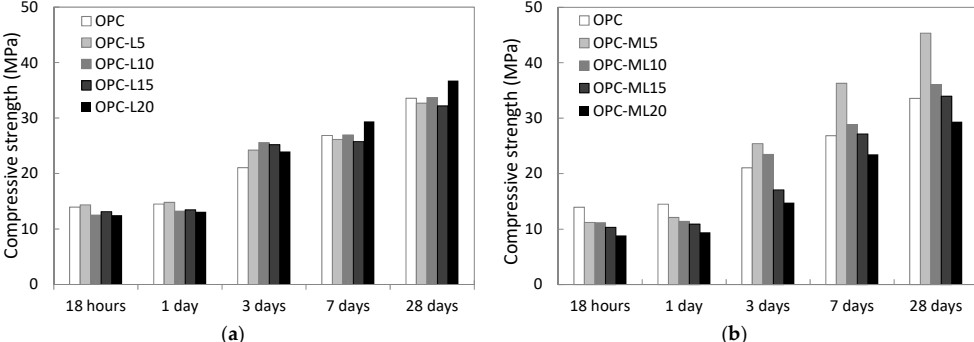

**Figure 6.** Compressive strength of cement mortar: (**a**) mortar using conventional LSSP and (**b**) mortar using modified LSSP.

It has been reported that the rheological properties are improved when limestone powder is replaced with cement. In addition, limestone powder is reported to be effective in reducing hydration heat and in increasing the initial strength. However, it is expected that both the long-term strength and the chlorine ion penetration resistance are decreased [32]. When the modified LSSP was replaced with cement at up to 10% of the cement weight, the initial (up to 3 days of aging) strength was low. On the other hand, the compressive strength of the mortar was as good as that of OPC after 28 days of aging, and a 5% substitution showed higher strength than those of OPC and conventional LSSP.

### 3.3. X-ray Diffraction Analysis

Figure 7 shows the results of the XRD analysis according to the cement substitution ratios of the conventional and modified LSSP after curing for 28 days. For the substitution with the conventional LSSP, the XRD analysis showed that the peak of calcium carbonate gradually increased and that the peak of calcium hydroxide decreased. This phenomenon has also been shown to reduce peak of ettringite and C–S–H. This might have been the cause of the initial strength reduction in the case of substitution with the conventional LSSP. In this case, however, it is believed that high-density filling by calcium carbonate increased the strength, and that the initial strength reduction was not significant.

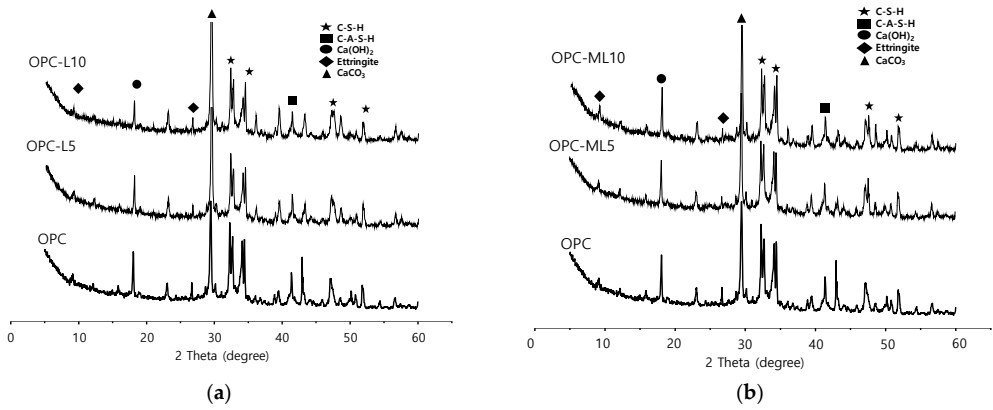

**Figure 7.** X-ray diffraction (XRD) patterns of samples with different contents: (**a**) conventional LSSP and (**b**) modified LSSP.

### 3.4. Thermogravimetric/Differential Thermal Analysis

Calcium carbonate, which is a major component of limestone powder, is mostly non-hydrated, and, thus, it is possible that the initial strength can be lowered when a substantial amount of LSSP is used. Figures 8 and 9 show the results of the TG-DTAs for different contents of conventional and modified LSSP, respectively. The temperature range of the cured specimens were measured at a maximum temperature of 900 °C. In Figure 8, comparison of the dehydration amounts at 450 °C shows that the amount of calcium hydroxide decreased with increasing LSSP content. In the case of the modified LSSP, Figure 9 shows that the reduction in the amount of calcium hydroxide was not significant, even when the substitution ratio increased. This suggests that the calcium acetate component on the surface of the modified LSSP was absorbed by tricalcium aluminate ($C_3A$) and converted into calcium hydroxide, thereby fixing the internal amount of calcium hydroxide. Calcium chloride and calcium nitrate are known to produce calcium aluminate hydrate combined with anion through hydration reaction with $C_3A$ in cement and to release calcium ions. When anions substitute $OH^-$ in the cement to produce hydrates, the order of anions affecting the hydration of cement is $OH^-$ < $Cl^-$ < $NO_3^-$ < $Br^-$ < acetate. Acetate ions are expected to produce hydrates of $C_3A$ and to produce calcium hydroxide. The amount of calcium hydroxide determined by TGA is shown in Table 3 [33–35].

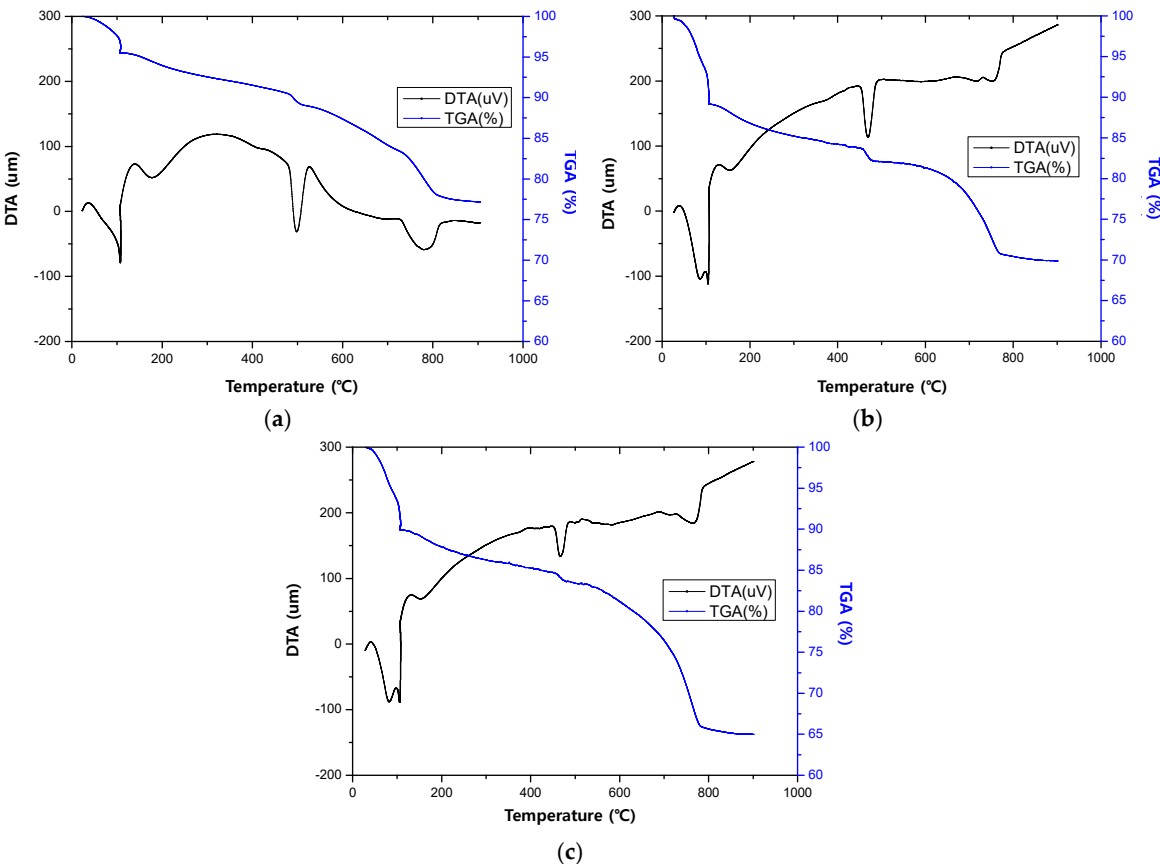

**Figure 8.** TG-DTA results for samples cured for 28 days: (**a**) OPC, (**b**) OPC with 5% conventional LSSP, and (**c**) OPC with 10% conventional LSSP.

**Table 3.** Quantification of the calcium hydroxide (28 days).

| Sample | OPC (Plain) | OPC-L5 | OPC-L10 | OPC-ML5 | OPC-ML10 |
|---|---|---|---|---|---|
| Ca(OH)$_2$ Weight content (%) | 2.33 | 1.01 | 0.985 | 1.771 | 1.876 |

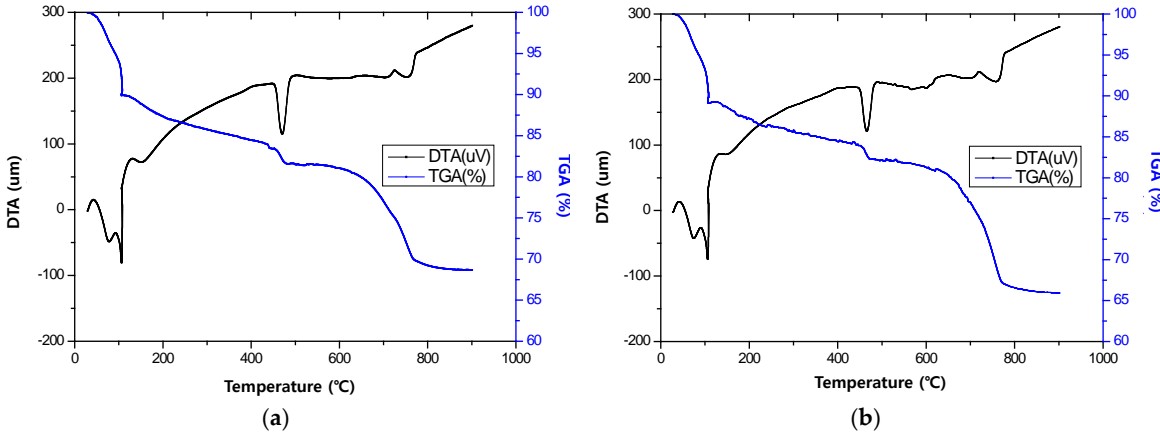

**Figure 9.** Results of the TG-DTA for samples cured for 28 days: (**a**) OPC with 5% modified LSSP and (**b**) OPC with 10% modified LSSP.

*3.5. Scanning Electron Microscopy Analysis*

Figure 10 shows the SEM images of samples with 5% substitution of conventional and modified LSSP, respectively. When reagent-grade calcium carbonate was mixed, it was observed that the calcium carbonate was held between the hydrates. Hydrates, such as ettringite, were formed on the surface, which enabled compact filling through physical filling and induced hydrate formation in the cement mortar.

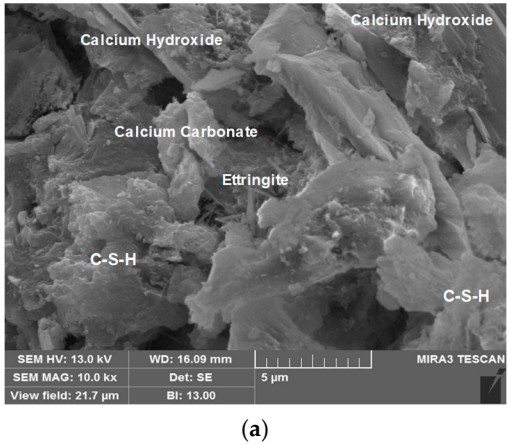
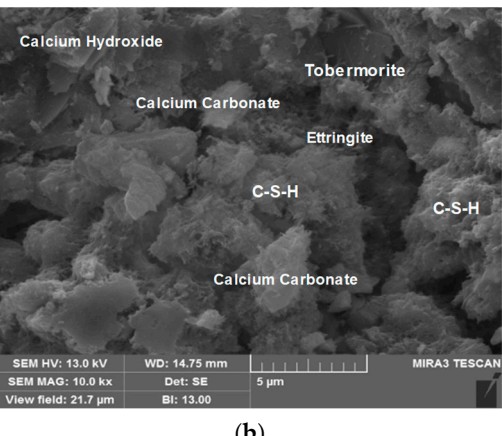

**Figure 10.** SEM images of samples: (**a**) 5% conventional LSSP (10,000×) and (**b**) 5% modified LSSP (10,000×).

## 4. Conclusions

The conclusions of this study can be summarized as follows.

Limestone powder is used as a raw material or admixture of cement. In this study, various properties of limestone sludge powder (LSSP) were evaluated by using recycled acetic acid (RAA), which is one of the industrial byproducts, as a cement admixture after surface modification of LSSP.

The surface of LSSP modified with RAA was converted into calcium acetate and had a large grain size. Calcium acetate has effects on mortar properties. The setting time of mortars with conventional LSSP was independent of the mixing ratios and was equal to that of ordinary type 1 Portland cement (OPC), whereas the initial compressive strength slightly increased. When conventional LSSP is used as a cement admixture material, it is possible that physical performance may deteriorate in the long term. Nevertheless, the use of modified LSSP with RAA can promote and continuously induce hydration in the long term, thereby improving the performance of the cement mortar. When the modified LSSP

was replaced with cement up to 10% of the cement weight, the initial (up to 3 days of aging) strength was low. On the other hand, the compressive strength of mortar was as good as that of OPC after 28 days of aging, and 5% substitution showed higher strength than those of OPC and conventional LSSP. Therefore, the use of cement admixture through modification of LSSP using RAA in this study is expected to be useful as a sustainable construction material.

**Author Contributions:** For research articles, author contributions are as follows; conceptualization, H.-S.R.; methodology and investigation, D.-M.K.; data curation, S.-H.S.; writing—original draft preparation, H.-S.R.; writing and editing, W.-J.P.; review and editing, S.-M.L. and W.-K.K.

**Acknowledgments:** This work was supported by the National Research Foundation of Korea (NRF) grant funded by the Korea government (no. NRF-2018R1D1A3B07045700).

**Conflicts of Interest:** The authors declare no conflict of interest. The founding sponsors had no role in the design of the study; in the collection, analyses, or interpretation of data; in the writing of the manuscript; and in the decision to publish the results.

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
