# Peer review of "Properties of Cement Mortar Using Limestone Sludge Powder Modified with Recycled Acetic Acid"

_sustainability, doi:10.3390/su11030879_

Reviewer 1 Report

It was a pleasure to read and review this article. The goal of this manuscript is attempting to characterize low-carbon cement using limestone sludge modified with recycled acetic acid. This experimental study is interesting and the subject fits within the scope of the journal. However, the reviewer would rather hesitate to recommend the acceptance of the manuscript with the current form to the journal of Sustainability. The authors are recommended to consider the following suggestions for reconsideration of the article.

1)      Figure 7 and related discussion: Based on the sustainable point of view, OPC-L20 (without additional treatment) is more promising compared to OPC-ML series.

2)      Line number 169, ‘decreased the amount of CaO’: Please discuss details regarding the amount of CaO since limestone has more CaO compared to OPC as shown in Table 1.

3)      Line numbers 30~33: Even though the authors mentioned that limestone powders are widely used for cement production, the authors also claimed that Canada & US allow to use the limestone powders up to 5% for Portland cement.

4)      Line numbers 31~35: It is difficult to find connections between two sentences.

5)      Line numbers 39~42: The English grammar of the sentence needs to be improved.

6)      Line numbers 70~71: Please explain the detailed procedure to get the results for Fig. 2, e.g., used system.

7)      Section 2.1: It is suggested for authors to provide particle size information of the raw materials including OPC, limestone powder.

8)      Line number 97: Please explain why the authors used a beam-shaped model just for compressive strength without bending test. And, the specimen size seems to be small. The authors need to be provided how many specimens tested per case.

9)      Table 2, wt%: This is not actually wt%. If the authors used the term percent, their sum should be 100.

10)  Line number 130, ‘no difference’: The authors needed to be used the expression as ‘no noticeable difference’ since the results are not the same.

11)  Section 3.3: The age information of the tested specimen for the XRD analysis needs to be provided.

12)  Reference 22~26: The reference seems to be redundant. The authors may want to use just one of reference.

Author Response

Thank you for your comments. Please find the attached.

Reviewer 2 Report

This manuscript aims to develop sustainable cement by using RAA modified limestone sludge (LSS). This is interesting and new. Proper test methods were used to characterize the hydrated paste using XRD, SEM, TG and mortar by compressive strength. However, some parts of the discussion need more explanations or references to support the statements.

Here are some technical comments:

Lines 70-71: “Figure 2 shows the XRD results…” should be Figure 2 shows the FTIR result for the RAA.

Figure 1: it would be better to plot the XRD results in 2D rather than 3D.

Line 133: What hydration reactions induced by calcium acetate?

Lines 134-135: This statement should be supported by literature.

Lines 189-190: this statement needs support.

Figure 11: Hydration products should be clearly labelled in the SEM images.

Line 226: The treatment of RAA to LSS doesn’t seem to increase the initial strength, as shown in Fig. 7.

Line 229-230: It needs support to state the long-term problem of cement containing limestone.    

Author Response

Thank you for your comments. Please fine the attached.

Reviewer 3 Report

Paper needs significant changes before it may be accepted. My comments are below:

1. Language: In general the language needs work. There are a few issues with grammar, spelling, phrasing etc. These make the work hard to understand and they need to be corrected. Please check the text carefully for language issues. A review from a native speaker will help.

2. How was the data in Table 1 obtained? Usually limestone has only about 56 % CaO (the rest is loss on ignition) when you test using XRF so this may be confusing. I think Table 1 needs LOI.

3. Figure 1: Change Lime Stone to Limestone (also in Figure 3). Is this figure needed? I am not sure this is telling you anything more than Table 1.

4. “Figure 2 shows the XRD results of CaCO3 and LSS.” This is incorrect, please check.

5. What does 60% RAA mean? Is this an aqueous solution? Please clarify. Later you say that 10% was used. Which is right?

6. What is the extent of reaction (1)?

7. Why cure underwater? You will have some leaching.

8. Were samples gold coated prior to SEM? For XRD and TGA, how were samples powdered?

9. L112: Calcium carbonation or calcium acetate?

10. There is some issue with the TGA. The decomposition is CaCO3 -> CaO + CO2. So, for 100 grams, you lose 44 grams CO2, which is a 44 % weight loss. You have a much higher weight loss in Fig. 5. Why? Am I missing something? Why show the DTA if it is not discussed?

11. Setting time: How does calcium acetate affect hydration? Can you compare to literature? Section 3.2.1 can be improved with references.

12. In general, some discussion about error/variability will help.

13. “At the substitution rate of 5%, the compressive strength increased, but the strength decreased by 10% or more for all specimens.” Confusing. Rephrase.

14. Do the conventional LSS and modified LSS have similar particle sizes? Can you show a curve of particle size distribution?

15. Figure 8: The red line under ettringite should be removed. Peak heights in general should not be used for quantitative purposes, and this should be noted.

16. “It is believed that the reduced amount of cement due to the substitution of the conventional LSS decreased the amount of CaO in the whole paste and consequently decreased the amount of calcium hydroxide.” Vague and wrong. If you replace cement with LS, calcium hydroxide is reduced because of dilution (there is lesser cement and calcium hydroxide is coming from cement hydration). Note that there is usually no CaO (free lime) in cement pastes; this is present as alite or belite.

17. CSH should be C-S-H.

18. “Although the amount of cement was reduced, there was active hydrate formation by the modified LSS.” You really cannot and should not make such conclusions from qualitative XRD.

19. TGA allows you to quantify calcium hydroxide amounts. Please do so. This is needed as such statements should not be made using qualitative data.

20. Figure 11: All of these images look exactly the same to me! I am not sure you can comment on surface characteristics from a fracture surface (which is inherently non-representative).

21. A detailed comparison to literature is really missing. Specifically something regarding what acetates do to cement hydration is needed here.

Author Response

(The authors gave the same response as above.)

Reviewer 4 Report

Title: Properties of cement mortar using limestone sludge modified with recycled acetic acid.

General comments

The paper reports on the substitution of Portland cement using Limestone Sludge (LSS). The LSS is used in two formats, one modified version and one non-modified version. The modification of the LSS was carried out using acetic acid. The manuscript is fairly well-written, but some sections are not clear and/or not fully explained. These are specified in the specific comments section. An annotated copy is also attached.

Specific comments

The issue of substitution is poorly defined/explained. It should be clear what is substituting what. In the abstract for example, it is not clear what is meant by “manufacturing low carbon cement is substituting limestone powders”. Is it the cement that is substituting limestone or vice versa? There are more examples of such confusion or lack of clarity on the issue substitution in the manuscript.

In the introduction, out of nowhere, the issue of polymers is introduced (Lines 35-36). When the authors state “physical performance of the polymer”, and they had not talked of any polymers prior to this, the reader is left wondering what polymer?

In section 2.1 on materials, it is not clearly specified whether LSS was in liquid or powder form. A sludge is typically a thick liquid. LSS Powder would better be abbreviated as LSSP.

In Figures 1 and 2, axis titles and/or units should be inserted where possible.

The reporting of compression strength is flawed.

Firstly, in the measurement of compressive strength from beam-shaped specimens, how was this accomplished?

Secondly, the strength increase referred to in lines 145-146 applies only during the early curing. The authors do not specify this.

Thirdly, substitution levels of 5% and 10% show better strength than control after day 1 for the modified LSS, yet that authors refer to after 3 and 28 days. This does not make sense.

The authors have attributed sustained presence of calcium hydroxide to conversion of calcium acetate by C3A into calcium hydroxide. Since the authors had reported earlier of increased strength at these substitution levels, it is also evident/possible that there is of enhancement of PC hydration, despite its substitution, as long as the levels of substitution are maintained at low levels of 5-10%. This angle of argument is not pursued or mentioned.

Figure 9(a) should have the same axis scale for TGA as the others – 9(b), 10(a) and 10(b).

SEM results are hardly discussed, yet there is a conclusion on this.

The conclusion in general are rather weak, especially the second conclusion which does not read very well.

It is uncommon in a journal paper to specify what each of the authors has done. There is nothing wrong with this, but usually reader does not want/need to know this. What is the benefit to the reader?

Author Response

Thank you for your comments. Please fine the attached.

Round  2

Reviewer 1 Report

The concerns raised by the reviewer were resolved in the revised manuscript.

Author Response

Thank you for your comments.
We had asked the native to correct the 2nd revised manuscript( English language and style are fine/minor spell check).

Reviewer 3 Report

Table 1 is confusing. Limestone normally has 40 % LOI, but you do not show any LOI for the limestone. The LSSP has 90.46 % CaCO3 and 51.50 % CaO? I would format consistently.

I think you removed XRD in Fig. 1 and are showing FT-IR. Can you explain how the FT-IR was done, the sample preparation, etc?

If 60 % of the RAA is solids, what is the rest? Please clearly state. 

Section 2.2: Please present details for SEM, TGA, XRD, particle size with references. Is stopping hydration by immersing in acetone for one day enough - what size of sample was used? Comment with references. 

Please clearly state that peak heights in XRD should not be used for quantitative purposes as they depend on numerous factors.

Can you explain how calcium hydroxide amount was determined with references? You cannot use two numbers, you normally need to use a tangential or horizontal baseline. 

What age are the calcium hydroxide numbers at? What are the values as a % of? Where is the value for the control OPC?

Why are different notations used? Table 2 has OPC-ML20 but Table 3 has Modified LSSP 10%.

You really need to present all details, its very hard to interpret this work otherwise. I have a hard time understanding the value of this work without the necessary details.

Author Response

Thank you for your detailed review comments. In addition, I feel sorry for the unclear expression and insufficient information in the manuscript. The answers to the review comments are summarized in the attached file. In addition, the 2nd revised manuscript was given to the native speaker in English.

Round  3

Reviewer 3 Report

I think the authors are not following my comments as they are not adequately addressed. Manuscript is still vague. 

Limestone has 44 % CaO and 36 % LOI. So what is the remaining 20%?

Please clearly state that peak heights in XRD should not be used for quantitative purposes as they depend on numerous factors. - I am asking you to add this statement to the text to keep it from being misleading. 

TGA -> So why are you still presenting masses at different temperatures? I do not follow. 

What age are the calcium hydroxide numbers at? What are the values as a % of? Where is the value for the control OPC? Why are different notations used? Table 2 has OPC-ML20 but Table 3 has Modified LSSP 10%. - You have not answered all parts of this question. Please reply in detail to these comments. Why do you have increasing calcium hydroxide as limestone replacement increases? That makes no sense.

Author Response

Thank you for your review comments. Please find the attached file.

Round  4

Reviewer 3 Report

You have still not responded to all parts of the question regarding TGA. As presented, I am not sure the TGA data makes sense.